# Mechanical Properties of Three Bamboo Species: Effect of External Climatic Conditions and Fungal Infestation in Laboratory Conditions

**Asier Elejoste** [1], **Juan Luis Osa** [2], **Alfonso Arevalillo-Roman** [3], **Arantxa Eceiza** [1], **Jose Miguel Abascal** [3], **Jose Miguel Rico-Martinez** [4], **Amaia Butron** [3] and **Cristina Peña-Rodriguez** [1,*]

1   'Materials + Technologies' Research Group (GMT), Chemical and Environmental Engineering Department, Faculty of Engineering, Gipuzkoa, University of the Basque Country (UPV/EHU), 20018 San Sebastián, Spain
2   Mechanical Engineering Department, Faculty of Engineering, Gipuzkoa, University of the Basque Country (UPV/EHU), 20600 Eibar, Spain
3   TECNALIA, Basque Research and Technology Alliance (BRTA), 20730 Azpeitia, Spain
4   Architecture Department, School of Architecture, University of the Basque Country (UPV/EHU), 20018 San Sebastián, Spain
*   Correspondence: cristina.pr@ehu.eus

**Abstract:** Bamboo is a material with good tensile and flexural resistance. As a construction material with structural capacity, using bamboo implies considerable environmental advantages in relation to other typical materials such as steel or concrete. For its correct implementation, it is necessary to define its mechanical properties and durability. Bamboo is susceptible to degradation due to the lack of natural toxins and thin walls, which means that shallow decomposition processes can imply appreciable reductions in its mechanical capacity. The main degrading agents considered in this study were beetles, termites, and xylophagous fungi. The aim of this study was to analyze the durability of three different species: DS, PA, and AA. Durability and mechanical tests results after 6 months of exposure to biotic and abiotic agents were compared with their original properties and chemical composition. In this study, durability was analyzed in two ways. Firstly, the loss of mass due to fungal infection was investigated. The results obtained were based on the standard EN 113 using the fungus CP. Secondly, bending and compressive strength was evaluated after a durability test according to the standard EN 335:2013 for the CU3.1 use class after a 6 month period in the city of Donostia/San Sebastian, Spain. The DS and AA varieties were rated as very durable CD1, while the PA variety is durable CD2, thus proving to be an attractive material for construction.

**Keywords:** bamboo; durability; bending strength; construction materials

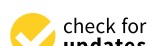



## 1. Introduction

During the last few decades, many research groups have been dedicated to develop and promote more sustainable materials for the construction sector. Timber has been one of the most commonly used sustainable building materials in Europe. However, in Asia, there is a widespread use of bamboo in construction. Even if both are lignocellulosic materials, their different internal structures imply distinct mechanical and structural properties. The interesting properties of bamboo are making it more and more popular in the world [1–5]. According to the American Bamboo Society, by 1988, it had more than 4000 uses [6]. However, its relatively high composition in starch, reducing sugars, and proteins makes it easily degradable by fungi in a hot and humid environment, which reduces its application value [7–9]. It generally has a low natural durability, and fungi can easily attack it during storage, transport, and final use. For applications, it is important to know its susceptibility [9].

Bamboo is not exclusive to Asia, as it grows naturally on all continents, and it is estimated that more than 22 million hectares in the world are intended for its cultivation [6],

with Brazil being the country with the greatest diversity of species [10,11]. However, in the Western world, it is still little known [12,13].

These plants are fast-growing giant herbs that have woody stems. The characteristics of each vary in size, growth habit, sun tolerance, soil moisture needs, and tolerance to cold and hot temperatures. The culm is a natural hierarchical cellular material that has good mechanical properties, including tensile and flexural strength, along the direction of its fibers. Since it is a functionally graded natural compound, the interfaces among its different ingredients, including fibers, parenchymal cells, and lignin matrix, can have a significant impact on its mechanical properties [14]. The hierarchical microstructure is due to the vascular bundles of the parenchyma matrix surrounded by support cellulose fibers. These fibers provide the main mechanical properties. In addition, cellulose fibers act as reinforcement to strengthen the lignin matrix, similar to polymeric matrix composites reinforced with fibers. This structure creates crystalline and amorphous regions within the microstructure, where linear glucose chains with hydrogen bonds form crystalline regions, while irregular hydrogen bonds create amorphous regions [15,16].

Better mechanical properties are observed along the direction of its fiber than along the transverse direction. The unique microstructural properties of natural bamboo with respect to its mechanical properties make it a suitable renewable material for composite materials in high-performance applications. Typically, the density is higher on the outside surface and decreases toward the inner layers of the cross-section of its wall [4,17–21]. Therefore, the outer layers of stalks are supposed to have better mechanical properties [21–23]. However, to date, no exhaustive and systematic studies in terms of density and geometry of the culm, including wall thickness, diameter, and height of the culm, have been developed.

Rot fungi inflict severe damage to bamboo and timber constructions and need expensive refurbishment measures [8,24]. Bamboo is more susceptible to decay than wood, due to the lack of natural toxins [25] and their typically thin walls, which means that shallow decay processes may involve appreciable reductions in their mechanical capacity. The significant environmental biotic pressure on the culm and its products necessitates protective measures to put in place. The number of insects that feed on bamboo is estimated to consist of at least 1200 species, of which degrading fungi account for more than 400 species [26].

There are mainly three causes of deterioration.

i.  First, certain beetles (*Hylotrupes bajulus* larvae) are attracted to the starch and lay their eggs inside the culm. After that, the eggs hatch and the larvae feed along the stem and eventually through the stem walls to escape, leaving small round or oval outlet holes. The attack speed is faster on fresh green bamboo (it is more susceptible); however, even when dry, it can be attacked in warm and humid climates, where the balance moisture content of bamboo on the outside (under cover) is usually higher than that in more temperate climates [27].

ii. Second, termites are little insects similar to ants that live in colonies and feed on plant material. They are also attracted to bamboo starch; however, unlike beetles, they have enzymes that allow them to break down cellulose. As they live in large colonies, they can cause short-term damage. There are two generic types of termites depending on their habitat: underground or dry wood. The former live in the ground (preferably moist), while the latter construct their nests in the wood itself or bamboo. Underground termites are translucent; thus, they build tunnels or find hidden paths to avoid sunlight [28].

iii. Third, xylophagous fungi cause rot. For the fungus to thrive, the culm must be relatively wet with at least 20% moisture, which essentially means that it must be exposed to rain or soil moisture [29]. During post-harvest processing, mold and micro-fungi cause severe damage, devaluating its economic potential [8,30–32].

As with wood, the most effective ways to protect bamboo from deterioration are drying before use and design or construction measures. It must remain dry, protecting it from wind and rain. Prolonged exposure to water should be avoided. This will prevent putrefaction and reduce the rate of xylophage attacks [33,34]. In addition, humidity can

also be detrimental with danger of collapse, as occurs in fast-growing wood species such as poplar [35].

For the use of bamboo in its original form as a construction material with structural capabilities, it is necessary to know its mechanical properties and durability. Given the differences between species, it is important to characterize each one properly. Accordingly, as with conifer wood (pine, fir, etc.), with bamboo, it is necessary to create comparative tables according to the species and use. There are several tables describing bamboo characteristics, but they do not describe it with its original form, instead evaluating other products made from bamboo, such as battens and particleboards [36,37].

The aim of this study is to (i) analyze the durability of bamboo using EN 113:2021 (fungi infestation in laboratory conditions), (ii) assess the effect of external climatic conditions following EN 350:2016, and (iii) compare the results for three different species: DS, PA, and AA. These three species, although they have similar applications, are physically very different: finishing and appearance, interior structure, density of fibers, provision of feeding ducts, etc. [38]. The results in flexural and compressive tests from small samples can be applied to the structural design of larger structural elements considering their use class.

## 2. Materials and Methods

### 2.1. Materials

The materials analyzed were the most commonly used bamboo species in the manufacture of small structures: DS (*Dendrocalamus strictus*), PA (*Phyllostachys aurea*), and AA (*Arundinaria amabailis*). The bamboos were supplied by Bambusa Importaciones y Proyectos, SL and underwent a stabilizing treatment with borax salts, which improved the natural durability and other properties such as fire resistance. The bamboo provider delivered culms obtained in the seventh harvest of the bamboo plant, which can be considered homogeneous in terms of morphological and mechanical properties. The preparation of samples storage and transportation was performed according to ISO standard 22157:2019 [39].

The origin of the different species, the climatological growth conditions, the density of culm wall ($\rho$), and the total or average density of each species ($\rho'$) are shown in Tables 1 and 2, respectively. Since the objective was to use culm in its original form as a structural material, the whole culm was used in all the tests.

**Table 1.** Origin and growth conditions of the three bamboo species.

| Species | Origin | Altitude (m) | Precipitation (mm) | Moisture Content (%) | Zone Temperature (°C) | | |
|---------|--------|--------------|--------------------|----------------------|-----------|------|------|
| | | | | | Average | Max. | Min. |
| DS | Kanchanaburi (Thailand) | 400–600 | 1060 | 57–81 | 27 | 30 | 25 |
| PA | Anji County, Huzhou, | 200–500 | 1543 | 71–80 | 18 | 28 | −3 |
| AA | Zhejiang (China) | 200–500 | 1543 | 71–80 | 18 | 28 | −3 |

**Table 2.** Different characteristics of the three bamboo species.

| Species | Description | Moisture Content (%) | Density (kg/m³) | | Height (m) | Diameter (mm) | Age (Years) |
|---------|-------------|----------------------|-----------------|-----------------|------------|---------------|-------------|
| | | | $\rho$ (Culm Wall) | $\rho'$ (Total) | | | |
| DS | Cylindrical and solid form | $6.60 \pm 0.70$ | 624 | 624 | 5–15 | 30–50 | 3–4 |
| PA | Cylindrical and hollow form | $5.47 \pm 0.61$ | 863 | 362 | 6–9 | 30–50 | 3–4 |
| AA | Cylindrical and hollow form | $5.72 \pm 0.15$ | 940 | 497 | 6–13 | 20–60 | 3–4 |



*2.2. Methods*

In this study, durability was analyzed in two ways. The first focused on the loss of mass due to fungal infection, while the second was performed on samples that were outdoors for 6 months. Both durability tests were performed on the three bamboo species along with *Pinus sylvestris* L. (PS) specimens that were used as control samples.

The mechanical behavior of bamboo, as a natural material, shows a wide dispersion in terms of mechanical properties. The statistical analysis of results improved the predictions from a safety standpoint. The testing standards take into account this scatter, defining a minimum number of samples according to the measured property. Therefore, five samples per bamboo specie were used on the durability tests, while 25 samples were required in the mechanical tests (bending and compression tests). Furthermore, the EN 1058:2010 standard calls for the fifth percentile in the mechanical test to guarantee the integrity of the structures.

### 2.2.1. Durability EN 113:2021

The screening test was based on EN 113:2021 [40]. The screening test based on that standard consists of placing previously weighed and sterilized specimens in contact with the basidiomycetes fungus *Coniophora puteana* (CP) for 16 weeks in a climatic chamber at 22 °C and a relative humidity of 70%. This fungus was selected because it is the most virulent of the four fungi that the standard contemplates within the procedure of standardized tests, achieving an average mass loss between 40% and 50% in untreated wild pine. According to the literature, CP is not only more virulent than other fungi, but the rotting it causes also affects more mechanical properties of wood than that caused by other fungi [41–43].

Once the 16 weeks of contact were completed, the specimens were removed, the adhered mycelium was cleaned, and the wet weight was obtained, the specimens were introduced into the stove at 103 °C for 18–24 h until obtaining a constant weight or the dry weight. Comparing the initial dry weight and the final dry weight, the percentage of mass loss was obtained, which was required to obtain the durability class.

The conclusions from this test were based on EN 350:2016 [44]. The norm allocates wood durability classes against basidiomycetes attack, and the criteria for the allocation depend on the average mass loss obtained as a percentage.

### 2.2.2. Durability EN 335:2013

The samples were prepared for a use class CU3.1 for a period of 6 months in accordance with the EN 335:2013 standard [45]. During this natural durability test, bamboo samples were exposed to biotic and abiotic agents in the city of Donostia/San Sebastián (43°18′34.92″ N and 2°0′34.2″ W) located on the east coast of the Gulf of Biscay.

Figure 1 shows the layout, where specimens were fixed vertically with cable ties, avoiding any contact with the ground. Thus, a water deposit accumulated in the culm, accelerating the putrefaction phases, and the study was performed in the most adverse conditions. The weather conditions collected by Euskalmet (Basque Metereology Agency) at Miramón weather station (43°18′34.92″ N and 2°0′34.2″ W) during the 6 months of exposure are summarized in Figure 2. After completion of the weathering period, the samples were used for mechanical tests to compare the results with those obtained in the previous characterization study [21].

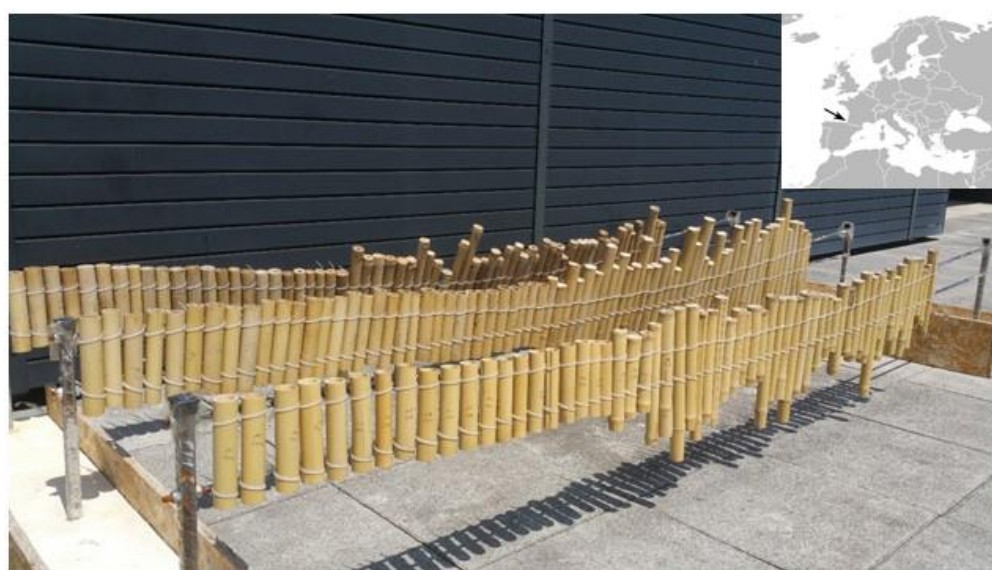

**Figure 1.** The 6 month weather exposure durability test and geographical location of the test (see map in upper right).

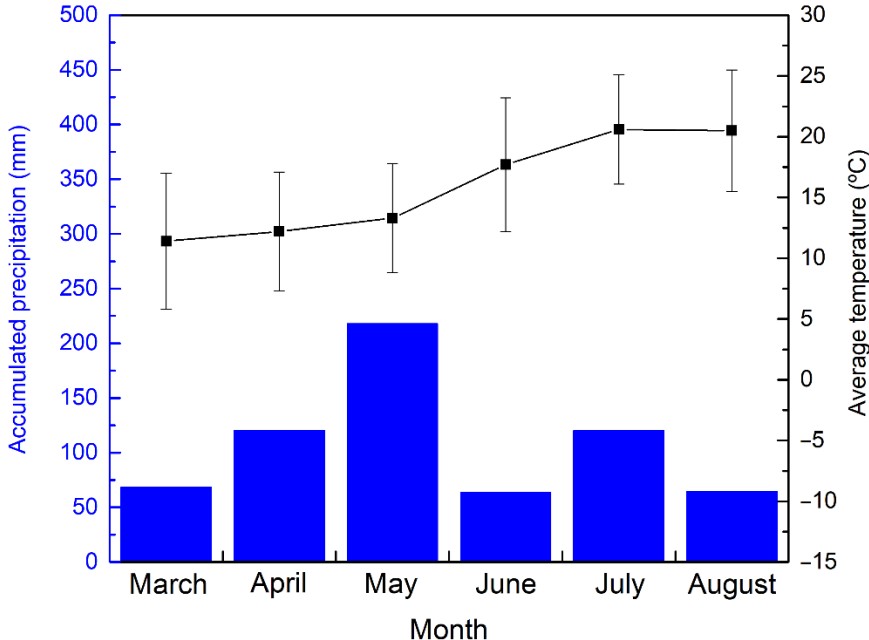

**Figure 2.** Graph showing the weather conditions during durability test EN 335 in Donostia/San Sebastian from March to August 2019.

### 2.2.3. Chemical Characterization of Bamboo

Different chemical components of the three bamboo species, including extractives and lignin, were experimentally measured. Firstly, samples were prepared using the TAPPI T 257 cm—85 and TAPPI T 264 cm—97 standards in order to measure the moisture content. For the extraction of extractives with ethanol toluene, TAPPI T 204 cm—97 standards were used. The benzene/ethanol mixture seemed to provide the most complete of all solvent-extractable substances in lignocellulosic materials. The percentage of insoluble acid lignin was obtained using the TAPPI T 222 cm—98.

### 2.2.4. Mechanical Tests

The mechanical properties of the three species before and after environmental exposition over 5 months (EN335:2013) were analyzed through bending and compression tests

(Figure 3). For this purpose, 25 specimens were used for each test type using two machines: Tinius Olsen H50 KN (displacement precision $\pm$ 0.026 mm) and Metrotec MS-7M 260 KN (load precision 0.5% between 50 KN and 260 KN; class 0.5). The characteristic values were obtained from the strain–stress curves obtained for each test. Maximum stresses were calculated using the load value for the first crack, not the maximum load. The speed used for the bending test was 3 mm/min, and the average test time was 51.58 s. In the compression test, the speed was 5 mm/min, with the average test time of 40.45 s.

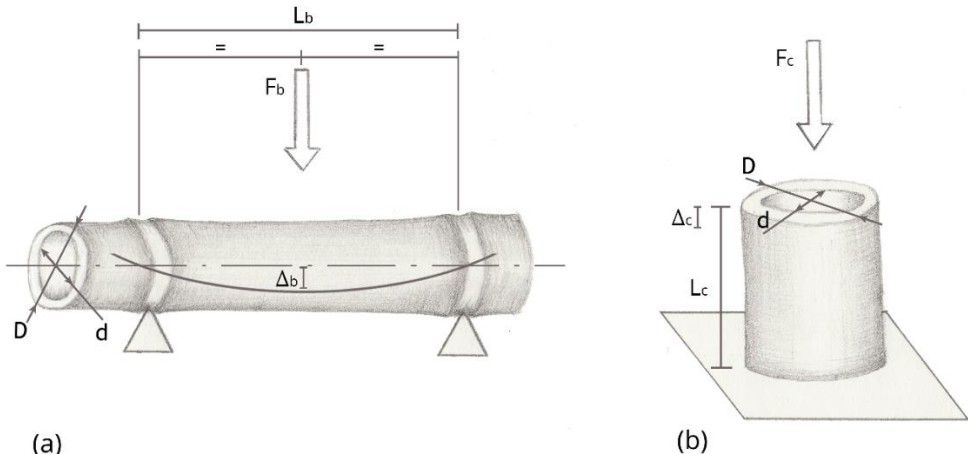

(a)                                                            (b)

**Figure 3.** Mechanical tests diagrams: (**a**) bending test; (**b**) compression test.

Figure 3 shows diagrams of the bending and compression tests and the related magnitudes according to ISO 22157, where $L_b$ is the clear span in flexural test (distance between nodes), $L_c$ is the sample length in the compression test, and D is the diameter of the cross-section of a piece of bamboo taken as the average of two perpendicular measurements made across opposite points on the outer surface. Measurement is usually made at the center of an internode region; d is the inside diameter or diameter of transverse hole, $F_b$ is the bending load applied at the mid-span, $F_c$ is the compressive load, $D_b$ is the mid-span deflection in bending, and $D_c$ is the shortening in compression.

Table 3 shows the dimensions of each type of bamboo studied in the tests. Because the DS culm is solid, it has no inside diameter.

**Table 3.** Dimensions of the specimens for mechanical tests. $L_b$ is the clear span in flexural test (distance between nodes), $L_c$ is the sample length in compression test, D is the diameter of the cross-section of a piece of bamboo taken as the average of two perpendicular measurements made across opposite points on the outer surface, and d is the inside diameter or diameter of transverse hole.

| Dimension (mm) | Bending | | | Compression | | |
|---|---|---|---|---|---|---|
| | DS | PA | AA | DS | PA | AA |
| $L_b$, $L_c$ | 180–360 | 130–350 | 170–420 | 180 | 150 | 180 |
| D | 27–41 | 27–33 | 31–38 | 28–39 | 27–33 | 28–38 |
| d | 0 | 17–31 | 21–29 | 0 | 20–26 | 21–28 |

Bending Strength and Stiffness Parallel to the Fibers

Given the dimensional variability of the samples, for this analysis, a three-point flexural test was chosen where the sample was supported on two simple supports (i.e., no moment restraint) situated below the two nodes, with the culm between them and the load applied at the mid-span (Figure 3).

The values obtained from the tests were bending strength ($\sigma_{bf}$, highest normal stress at fracture) and modulus of elasticity in bending ($E_b$) (Equation (2)). These were obtained following the formulae for the moment of inertia I (Equation (3)), section modulus *W*

(Equation (4)), and normal stresses $\sigma_b$ (Equation (1)) in a circular hollow section in bending, and the bending moment under a load in a three-point bending setup $M$.

$$\sigma_b = \frac{M}{W} = \frac{F_b L_b/4}{2I/D}, \tag{1}$$

$$E_b = \frac{(F_{b2} - F_{b1})L_b^3}{48(\Delta_{b2} - \Delta_{b1})I}, \tag{2}$$

$$I = \frac{\pi}{64}\left(D^4 - d^4\right), \tag{3}$$

$$W = \frac{\pi}{32D}\left(D^4 - d^4\right), \tag{4}$$

where $\sigma_b$ is the normal stress at the most distant fibers from the centroid of the circular hollow section (outer fibers), $M$ is the applied bending moment, $W$ is the section modulus of the outer fibers of a circular hollow section, $I$ is the moment of inertia of a circular hollow section, $E_b$ is the modulus of elasticity in bending, $F_{b1}$ is 10% of the maximum load and $F_{b2}$ is 40%, and $D_{b1}$ and $D_{b2}$ are deflections, where $D_{b1}$ is caused by force $F_{b1}$, and $D_{b2}$ is caused by $F_{b2}$. Measurements were taken according to the ISO 22157 standard.

Compression Strength and Stiffness Parallel to the Fibers

In the compression test, the sample rests vertically on a fixed base (Figure 3, right). The compression load depends on the upper loading platen located over the top end of the sample. The length of the specimens was 180 mm for the case of DS and AA, and 150 mm for PA. This difference was due to the shorter internode length in the case of PA. Since bamboo is a natural product, the diameter and thickness of the samples were variable; the diameter of DS ranged between 28 and 39 mm, the diameter of PA ranged between 27 and 33 mm, and the diameter of AA ranged between 28 and 38 mm (Table 3).

The values obtained were compression strength parallel to the fibers ($\sigma_{cf}$) (Equation (5)) and modulus of elasticity in compression parallel to the fibers, $E_c$ (Equation (7)). These were obtained considering the formula for the cross-sectional area $A$ (Equation (6)).

$$\sigma_{cf} = \frac{F_{c\ ult}}{A}, \tag{5}$$

$$A = \pi\left(\frac{D}{2}\right)^2 - \pi\left(\frac{d}{2}\right)^2 A = \frac{\pi}{4}\left(D^2 - d^2\right), \tag{6}$$

$$E_c = \frac{(F_{c2} - F_{c1})L_c}{(\Delta_{c2} - \Delta_{c1})A}, \tag{7}$$

where $\sigma_{cf}$ is the normal stress parallel to the fibers, $A$ is the cross-sectional area, $E_c$ is the modulus of elasticity in compression parallel to the fibers, $F_{c\ ult}$ is the maximum load at which the specimen fails, $F_{c1}$ is 10% of the maximum load and $F_{c2}$ is 40%, and $D_{c1}$ and $D_{c2}$ are shortening deformations, where $D_{c1}$ is caused by force $F_{c1}$, and $D_{c2}$ is caused by $F_{c2}$. Measurements were taken according to the ISO 22157 standard.

2.2.5. Density

Densities were calculated in two ways. First, the density of the culm wall ($\rho$) (Equation (8)) was calculated, taking the total mass of the bamboo and the volume of the wall. Second, the total or average density ($\rho'$) (Equation (9)) was calculated, taking the total mass and the total volume, including the hollow interior volume.

$$\rho = \frac{4m}{\pi(D^2 - d^2)L}, \tag{8}$$

$$\rho' = \frac{4m}{\pi D^2 L}, \tag{9}$$

where $d$ is the inside diameter of bamboo (mm), $D$ is the outer diameter (mm), $L$ is the length (mm), and m is the measured mass (g) of the specimen. Measurements were taken according to the ISO 22157 standard.

## 3. Results and Discussion

### 3.1. Durability against Fungus Basidiomycetes

The obtained durability classes (Table 4) depend on the mean quantified mass loss. After the results obtained from the screening test based on EN 113:2021, the durability classes were identified according to EN 350:2016. As noted, DS and AA varieties were classified as very durable DC1, while the PA variety was classified as durable DC2.

**Table 4.** Mean mass loss (ML$_1$) for DS, PA, and AA species after the durability test according to standard EN 350:2016 and mean mass loss (ML$_2$) according to standard EN335 CU3.1.

| Species | EN 350:2016 | | | EN335 CU3.1 |
| | ML$_1$ (%) | Durability Class | Description | ML$_2$ (%) |
|---|---|---|---|---|
| DS | 3.95 | DC1 | Very durable (ML ≤ 5) | 15.28 |
| PA | 5.65 | DC2 | Durable (5 < ML ≤ 10) | 10.15 |
| AA | 3.02 | DC1 | Very durable (ML ≤ 5) | 7.85 |
| PS | 56.71 | DC5 | No durable (30 < ML) | ——- |

Figure 4 also includes the mass loss in the weathered specimens according to EN 335. Comparing the two durability tests, it can be seen that the weathered specimens lost two or three times as much mass as the fungus-rotted specimens. This difference is due to the fact that, in EN 335, the specimens are exposed to the environment for a longer period of time (6 months compared to 4 months in EN 113:2021). In addition, several microorganisms from the environment are also involved in the decomposition of the bamboo, whereas, in EN 113:2021, the fungus CP is the only one responsible for the loss of mass. In both cases, the most important mass loss occurs inside the culm because it has less fiber density and is more likely to be attacked by external agents [38,46].

A decisive factor in the durability of plants is their chemical composition in terms of the amount of extractive substances, whereby the less the plant has, the less durable it will be [41]. Comparing the results of the toluene/ethanol extractive (Table 5), it can be stated that the values were between 10% and 15%, which would imply a good durability for these three species, especially compared to the PS control samples (0.90%). Extractive compounds can be classified into three subgroups according to their chemical composition: phenolic aromatic compounds such as tannins and lignins, aliphatic compounds (fats and waxes), and terpenes and terpenoids. While aliphatic compounds can act as surfactants that limit the adhesion of fungi to the wood surface, phenolics have rather a direct effect on the physiology of fungi [47]. In this research, differences were observed. The species with the highest percentage of extractives was PA, and, according to the durability results (Table 4), it was the least durable species. This may be due to the fact that, in the determination of extractives, toluene/ethanol was used in the process. In these solvents, sugars can also dissolve, which worsens the durability, invalidating the results obtained using this method.

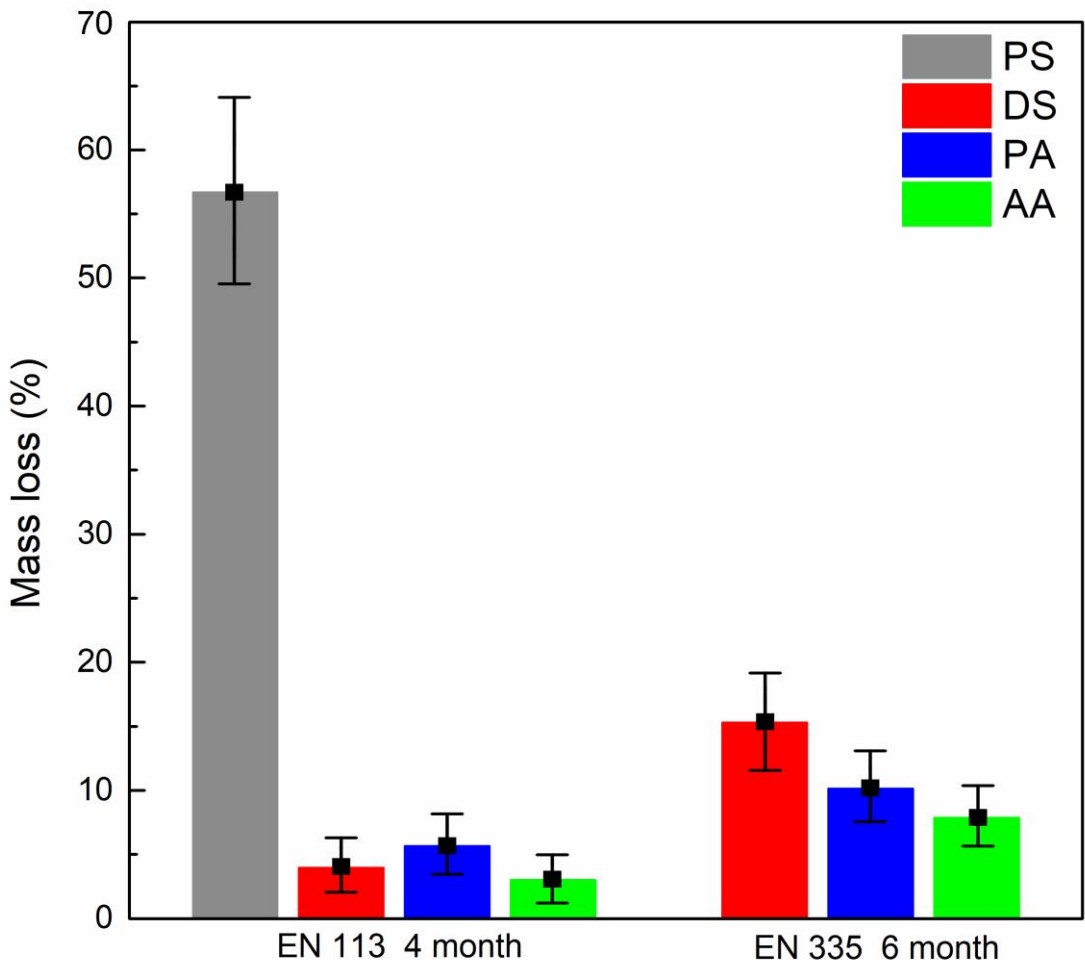

**Figure 4.** Representation of the mass loss results EN 113 and EN 335 of the bamboo specimens (DS, PA and AA) and of the control sample (PS).

**Table 5.** Chemical composition of DS, PA, and AA bamboos.

| Species | Composition | |
|---|---|---|
| | Extractives (%) | Lignin (%) |
| DS | 12.12 ± 0.05 | 23.80 ± 5.50 |
| PA | 14.71 ± 1.46 | 18.37 ± 1.55 |
| AA | 9.70 ± 1.34 | 28.85 ± 3.62 |
| PS [48] | 0.90 | 25.4 ± 8 |
| Bibliography [22,49–52] | 0.91–10.91 | 22.66–24.11 |

The durability of lignocellulosic materials was also related to the lignin content [53,54]. Species AA and DS had a lignin content of 29% and 24%, respectively, whereas this content was lower for species PA (19%), in good agreement with the durability of these species (Table 4). The highest lignin content was obtained with the AA species, coinciding with the bamboo that showed the highest durability according to the EN 350:2016 standard and the literature [42,55]. In fact, several studies have indicated that the physical and chemical properties of lignin serve as a barrier against pest and pathogen invasion [53,56]. Indeed, lignification is a mechanism of disease resistance in plants. During defense responses, accumulation of lignin or lignin-like phenolic compounds has been shown to occur in a variety of plant–microbe interactions [53,54].

### 3.2. Mechanical Properties

Stress and elasticity moduli of bamboos in fracture were measured under flexural and compression loads before and after 6 months of weathering in environmental conditions (EN335 standard). Average values and values of the fifth percentile are shown in Table 6. In order to increase safety due to the natural dispersion of the properties of natural materials, it is customary in the design of timber structures to use the values corresponding to the fifth percentile instead of the average values (EN 1058:2010 [57]). The same approach was adopted here.

**Table 6.** Mechanical properties and density of DS, PA, and AA. Initial values and values after 6 months weathering are included.

| | | | DS | | PA | | AA | |
|---|---|---|---|---|---|---|---|---|
| | | Time (Months) | 0 | 6 | 0 | 6 | 0 | 6 |
| Bending | Average | $\sigma_{bf}$ (MPa) | 90.3 ± 27.2 | 77.5 ± 16.6 | 40.5 ± 7.0 | 50.6 ± 25 | 55.3 ± 7.7 | 52.7 ± 11.5 |
| | | $E_b$ (MPa) | 3234 ± 2181 | 3932 ± 2010 | 4091 ± 1898 | 2960 ± 2427 | 6689.5 ± 2454.7 | 4846 ± 2116 |
| | 5th percentile | $\sigma_{bf}$ (MPa) | 58.5 | 52.2 | 31.6 | 30.1 | 45.1 | 36.3 |
| | | $E_b$ (MPa) | 921 | 1192 | 2042 | 1185 | 3039 | 1894 |
| Compression | Average | $\sigma_{cf}$ (MPa) | 50.4 ± 10.5 | 29.5 ± 7.7 | 71.4 ± 11.5 | 66.7 ± 8.7 | 78.2 ± 10.8 | 71 ± 13.2 |
| | | $E_c$ (MPa) | 4249 ± 1180 | 2674 ± 822 | 6555 ± 1605 | 5895 ± 1674 | 6645 ± 1921 | 6577 ± 1713 |
| | 5th percentile | $\sigma_{cf}$ (MPa) | 36.6 | 20 | 50.0 | 54.9 | 61.1 | 46.7 |
| | | $E_c$ (MPa) | 2454 | 1604 | 3818 | 4105 | 4513 | 3943 |
| Densities | | $\rho$ (kg/m$^3$) | 624.45 | 529 | 862.55 | 775 | 939.85 | 866 |
| | | $\rho'$ (kg/m$^3$) | 624.45 | 529 | 362.00 | 302 | 497.00 | 387 |

### 3.2.1. Bending

DS bamboo showed a noticeable reduction in the mean value of bending strength after the 6 month weathering durability test (Table 6). The decrease was smaller (15%) at the fifth percentile value. This was probably due to a degradation of the culm core. In the case of PA bamboo, a reduction in strength of less than 5% was observed at the fifth percentile, and the mean value was even higher. Although it suffered a 10% loss in mass, the strength was practically the same after 6 months.

In the case of AA bamboo, the mean values did not vary substantially after 6 months of aging, and the fifth percentile was significantly lower (20% decrease), making it the species most affected by weathering. Due to the morphology of this species, the inner part of the culm wall is almost free of fibers, which causes it to lose most of the mass in that inner part of the culm wall [38]. In the absence of affected fibers, this loss of mass should not be significant in terms of strength. However, it is important in the bending test, as this inner part of the bamboo helps to maintain the tubular shape of the culm. As for the stiffness analysis, DS did not appear as affected (it even appears to have improved). In the case of PA and AA, there was a significant reduction in the stiffness average value: over 26% in the case of PA and 27.5% in the case of AA.

In summary, degradation reduced the flexural strength in the fifth percentile in the three types of bamboo (DS 10%, PA 5%, and AA 20%), as well as the stiffness fifth percentile

in hollow culms (PA 42% and AA 37.6%). This is due to the morphology of hollow culm, where, although the interior does not provide structural capacity [38], it helps to maintain the tubular form in bending tests, a problem that does not occur with the solid species.

Figure 5 shows images of the bending test of the three species. DS (Figure 5a) was the most heavily loaded, but had a full inner section. The fibers in the bending tensile zone broke locally when the maximum shear stress limit was exceeded. The fact that DS has a solid top gives it an advantage over AA and PA, which have a hollow top. The fracture of the PA (Figure 5b) indicates that, once the test is finished and the load is removed, the PA returns to its shape in such a way that it is not possible to see with the naked eye the cracks generated during the test. Furthermore, it can be seen how the external agents were only able to attack the epidermis where the flange was holding the bamboo during the durability test.

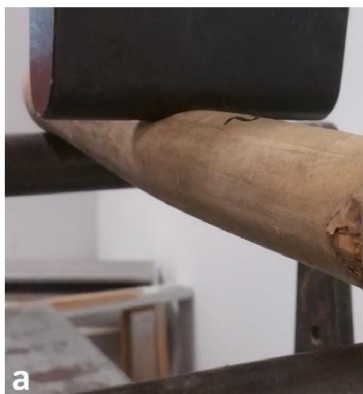 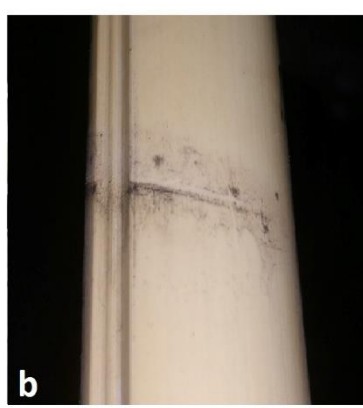 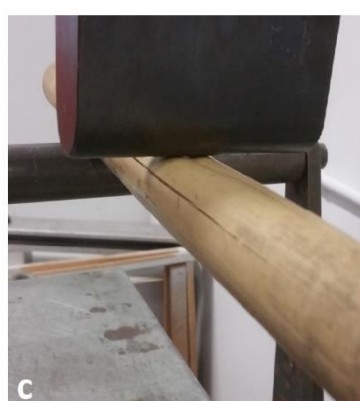

**Figure 5.** Pictures of fracture after bending test for (**a**) DS, (**b**) PA, and (**c**) AA.

The latter (Figure 5c) have a lateral flaring in the area where the clamp is applied, which results in bending in the plane of the cross-section. With reduced strength in the plane transverse to the direction of the fibers, longitudinal cracks appear along the culm [21]. The AA and PA diagrams in Figure 6 have the shape of a sawtooth. During the test, the load increased, and, at the initiation of a crack, a redistribution of stresses occurs until other fibers were mobilized, and the load increased again until the generation of the next crack began, generating a new sawtooth in the graph (Figure 6).

The fracture toughness of the materials was also analyzed in relation to their density (Figure 7), i.e., the specific strength, in order to compare these results with those obtained in the previous study [21], in which the same mechanical tests were carried out before the natural durability test.

Results indicated that the DS showed the highest fracture stresses at 6 months of testing. However, the results are misleading, as the fracture mechanism is different having a solid culm. AA and PA with hollow culms were comparable, as they have similar cross-sections and fractures. At similar densities, AA had a higher strength than PA. Figure 7 shows the dispersion of the results. Although it appeared in all three species, the DS results showed the greatest dispersion. This variability offers little certainty in the design of structural elements for future applications. The trend line of the PA is surprising, as it was observed that a higher density led to a lower strength. The loss of mass occurred in the inner part and, in the case of PA, upon losing the inner part, which was the least dense, the overall density increased. This inner part is necessary to maintain the tubular shape. Therefore, the loss of mass of the inner part increases the density of the culm and, in turn, decreases its resistance to bending.

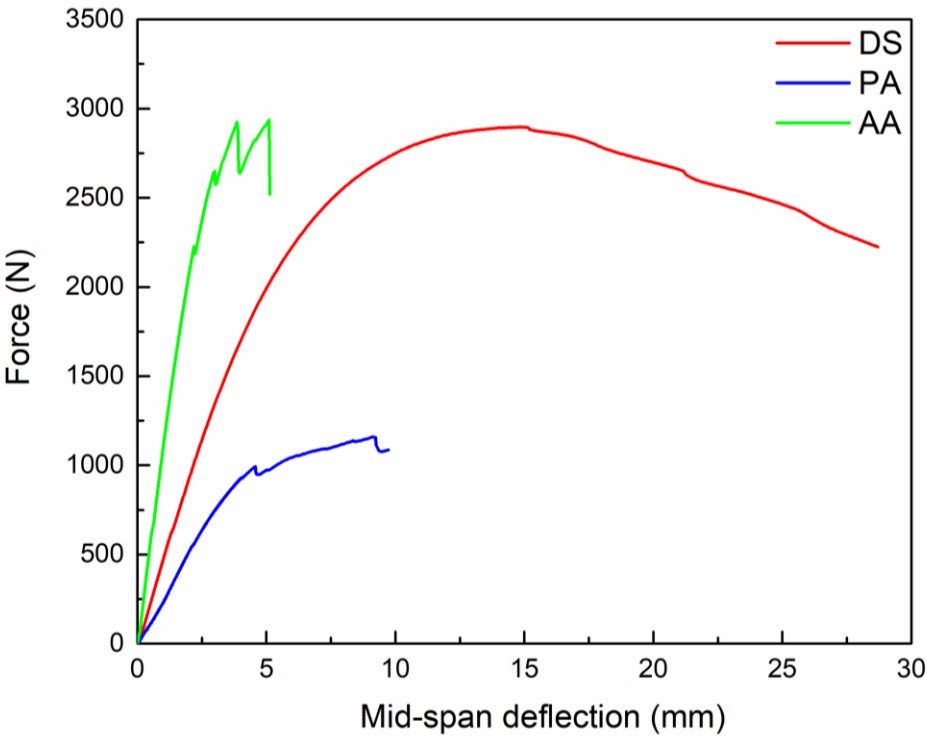

**Figure 6.** Bending load–displacement graphs ($F_b$–$D_b$) for the three species at 6 months.

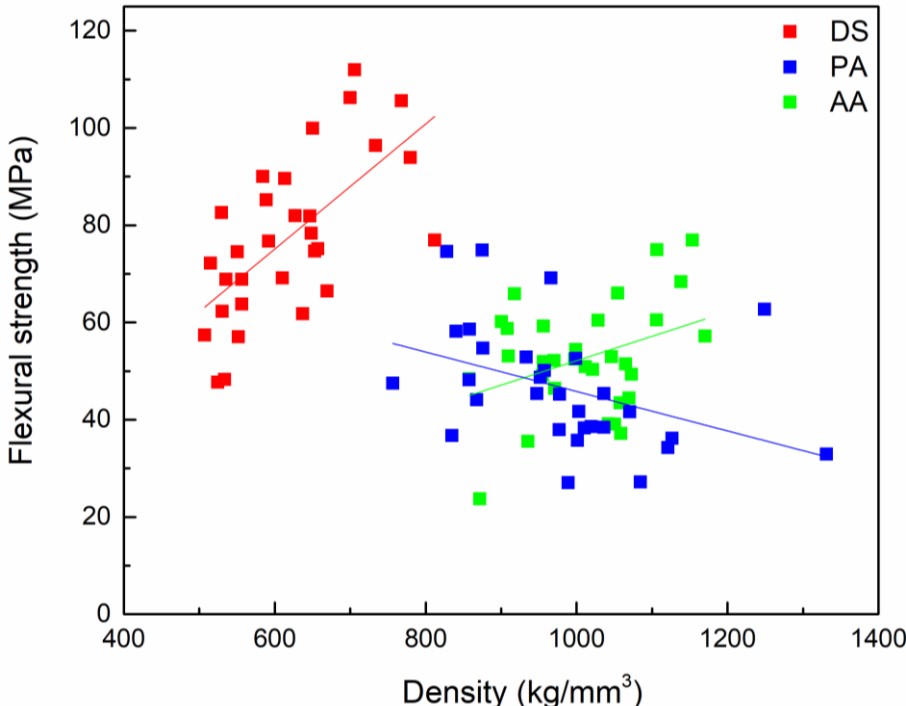

**Figure 7.** Graph showing relation of flexural fracture stresses with density after 6 months of aging. The lines in the graph represent the linear regression of the values obtained for each species.

### 3.2.2. Compression

The results of the compression test of the specimens obtained after 6 months of exposure to the environment are presented in Table 6. The values of compressive strength, stiffness, and density of each species showed a certain variability, as already detected in other tests [21]. The most variable in terms of strength and stiffness was the AA species. In terms of density, the variation in all species was similar. The higher variability of the

AA species means that, although its average strength was the highest, the value of the fifth percentile, i.e., that used in the structural calculations, was not. In contrast, this was true in the 0 month tests [21].

The density values indicated that the AA species suffered less loss in the culm wall, around 8%, which also resulted in a loss of strength of less than 10% and a loss of stiffness of around 1%. In this case, analyzing the fifth percentile, the greater variability of the samples evaluated at 6 months suggests that the values were considerably reduced, but this was due to the variability of the material and not to the loss of mass.

The PA species suffered losses in wall density of around 10%, which also resulted in a slight loss of strength and stiffness. Analyzing the average values, the loss in strength was less than 7% and the loss in stiffness was less than 10%. Furthermore, when analyzing the fifth percentile value, the highest variability of the samples analyzed was found in the samples obtained at 6 months.

After 6 months of weathering, although all species lost density in their culm wall, the most affected by this compression test was the solid species (DS). Compared to the initial results, the DS species suffered a culm wall density loss of about 15%, which resulted in strength losses of about 45% and stiffness losses of about 35%. These are very high losses, probably due to the very morphology of bamboo, where most of the mass loss in hollow species occurs in the interior, which is easier to attack due to the lower concentration of fibers [38]. On the other hand, in the case of DS, the most affected area was the outside, with its consequent loss of mechanical properties.

Figure 8 shows an example of the load–displacement diagrams obtained in the compression tests. Figure 9 presents images of the fractures of these three species. In the three types of bamboo studied, DS, PA, and AA, the response to the compression test was continuous, with no sawtooth patterns appearing as in the bending tests.

In the case of DS, a bamboo with a solid culm, collapse appeared at the weakest point of the compression zone. This is where the longitudinal fibers began to separate under shear stresses. Once the separation started, the fracture extended continuously in the transverse plane.

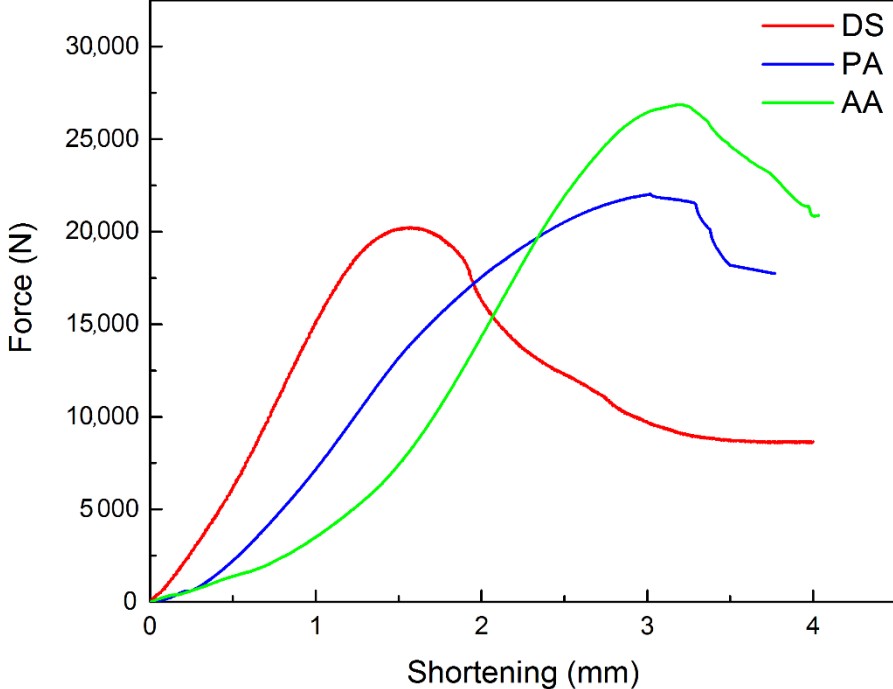

**Figure 8.** Compression load–displacement graphs ($F_c$–$D_c$) for the three species at 6 months.

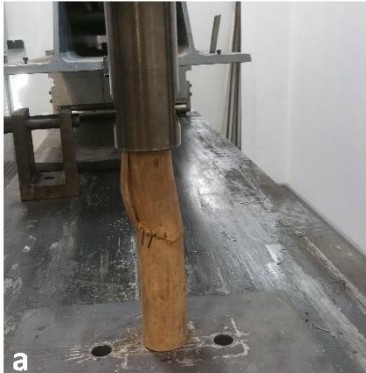
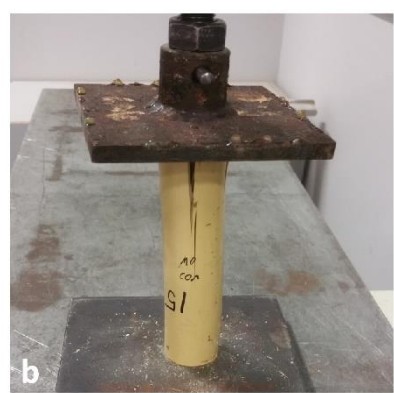
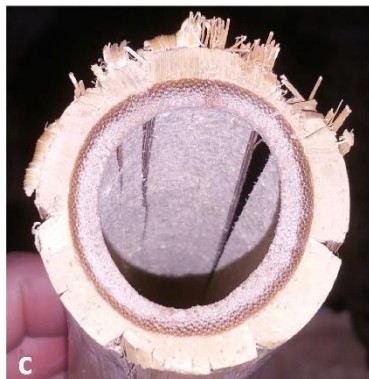

**Figure 9.** Pictures of specimen fractures after compression tests: (**a**) DS, (**b**) PA, and (**c**) AA.

As PA and AA have a hollow circular culm, their geometry is more optimal for resisting compression. When the shear stress of the material holding the fibers together was overcome, cracks appeared in the center of the culm. The gradual growth of the crack resulted in a continuous reduction in load on the graph. The cracks appeared mainly at the upper and lower ends of the specimen, together with longitudinal cracks arising in the center of the culm [21].

Figure 10 shows the relationship between compressive fracture toughness and density for the three types of bamboo. The graph shows that the density of the culm wall had a direct relationship with its compressive strength; a higher density of the wall led to a higher fracture stress and stiffness in compression. This direct relationship between wall density, strength, and stiffness also appeared at 0 months [21]. Analyzing Table 6, it can also be deduced that density had a similar influence on the modulus of elasticity: a higher density made the material stiffer. Figure 10 also shows the dispersion of material properties.

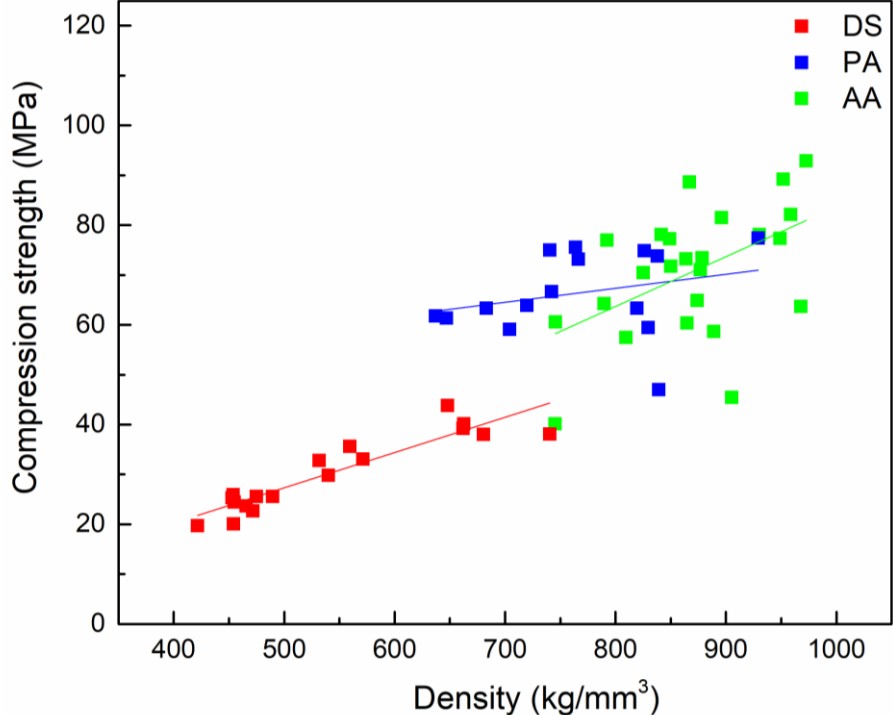

**Figure 10.** Graph showing relationship of compressive fracture stresses with density after 6 months ageing. The lines in the graph represent the linear regression of the values obtained for each species.

## 4. Conclusions

In order to use bamboo species safely and efficiently, it is necessary to carry out durability tests and evaluate the mechanical properties after six months of exposure to biotic and abiotic agents. The results obtained were contrasted with a previous study of characterization of these three same species [21], in order to observe differences in their performance.

The DS and AA bamboo varieties were found to be very durable CD1, while the PA variety was durable CD2.

Bending strength values of DS showed a reduction in the fifth percentile of about 10% at 6 months. For PA, the reduction was less than 5%, and, for AA, there was a reduction of about 20%. On the other hand, regarding the bending stiffness values, DS did not appear as affected, while a reduction in stiffness of more than 26% was observed for PA, with almost 27% for AA. The DS withstood the highest bending loads, as it had a solid culm section.

The fracture toughness of the materials was also analyzed in relation to their density. In general, a direct relationship was observed between the density of the culm wall and its strength and stiffness. A denser wall led to higher strength and stiffness. The DS results showed the greatest dispersion and, therefore, offer little certainty for the design of the structural elements. For PA, it was observed that a higher density led to lower strength, due to the fact that the loss of mass occurred in the internal part, which was the least dense, such that the overall density increased while the flexural strength decreased.

In compression, the most variable in terms of strength and stiffness was the AA species, which is why it also had the lowest fifth percentile value. In the case of PA, the average loss in strength was less than 7%, and the average loss in stiffness was less than 10%; however, if the fifth percentile was analyzed, the higher variability of the samples at 0 months resulted in slightly higher values at 6 months. After the 6 month weathering period, although all species lost wall density, the most affected was the solid species (DS) with losses in culm wall density, strength, and stiffness of 15%, 45%, and 35%, respectively.

Thus, it can be concluded that most of the mass loss in hollow species occurred on the inside. In the case of DS, the most affected area was the outside, where fibers were present in greater quantity, with a consequent loss of mechanical properties. As PA and AA have a hollow circular culm, their geometry was more optimal to resist compression. Accordingly, as in the case of bending, a higher wall density led to a higher fracture stress in compression. Therefore, morphology is a critical factor in the durability of bamboo. In hollow species, although the inner part does not provide structural capacity, it helps to maintain the tubular shape in bending tests, whereas this is not the case for solid species.

This work focused on three bamboo species. The same methodology should be applied on other species in order to enlarge the available comparative data. Protective treatments for increasing durability should be further tested to evaluate their efficiency in preventing the loss of mechanical properties.

**Author Contributions:** Conceptualization, C.P.-R., A.B. and A.E. (Asier Elejoste); methodology A.B., J.M.A., A.A.-R. and J.M.R.-M.; software, A.E. (Asier Elejoste) and J.L.O.; validation, C.P.-R. and A.B.; formal analysis, A.E. (Asier Elejoste), J.M.R.-M and J.M.A.; investigation, A.E. (Asier Elejoste); resources, A.E. (Arantxa Eceiza), C.P.-R., A.B.; data curation, J.M.A. and J.M.R.-M.; writing—original draft preparation, A.E. (Asier Elejoste); writing—review and editing, J.L.O., J.M.R.-M., A.B., C.P.-R. and A.E. (Arantxa Eceiza); visualization, C.P.-R. and A.B.; supervision, A.B. and C.P.-R.; project administration, C.P.-R. and A.B.; funding acquisition, C.P.-R., A.B. and A.E. (Arantxa Eceiza). All authors have read and agreed to the published version of the manuscript.

**Funding:** Financial support from the Basque Country Government in the frame of Grupos Consolidados (IT-1690-22).

**Acknowledgments:** The authors thank the Circular Economy University, Company Classroom (Faculty of Engineering Gipuzkoa, UPV/EHU, Provincial Council of Gipuzkoa). Moreover, the authors are grateful to the Macrobehavior–Mesostructure–Nanotechnology SGIker unit of UPV/EHU.

**Conflicts of Interest:** The authors declare no conflict of interest.

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
