# Peer review of "Mechanical Properties of Three Bamboo Species: Effect of External Climatic Conditions and Fungal Infestation in Laboratory Conditions"

_forests, doi:10.3390/f13122084_

Round 1

Reviewer 1 Report

This manuscript is informative and provides readers with a fresh insight into the durability of bamboo. After careful consideration, I believe this manuscript would be acceptable for publication after addressing the following comments:

(1) The abstract lacks a proper closing sentence. the implication of this research should be discussed in one sentence at the end of the abstract.

(2) The seventh paragraph In the introduction has only two sentences. This paragraph should discuss prevalent drying defects in bamboo. I also recommend authors compare drying imperfections such as internal checks in bamboo, and fast-growing wood species like poplar. The following paper on drying poplar is useful for comparison.

Rahimi, S., Faezipour, M., & Tarmian, A. (2011). Drying of internal-check prone poplar lumber using three different conventional kiln drying schedules. Journal of the Indian Academy of Wood Science8(1), 6-10.

(3) The critical missing part of this research is its practical application. I suggest authors explain how the results from small samples are applicable to big sizes (real-life problem solving).

(4) Conclusion is prolonged and resembles the abstract.

(5) Research limitations and recommendations for future studies should be provided.

Reviewer 2 Report

General comment:

Thank you for the opportunity to review the manuscript. The authors of this article the durability and mechanical characteristics of three different species of bamboo: DS (Dendrocalamus Strictus), PA (Phyllostachys aurea) and AA (Arundinaria amabailis). This remains an important and original research topic. I appreciate the work of the authors. In general, although the central idea of the article is of great interest, some fundamental aspects need to be reviewed and addressed to improve the scientific value of this research.

Title: It is not clear if the research work is only on abiotic agents or even abiotic "weathering effect" is very generic and not very impactful. I suggest the authors to change the title giving a more scientific approach.

Keywords: Words from the title should not be used as keywords. I suggest to modify, reducing the number and the means

References: Bibliographic references could be improved. I believe that authors should look for further bibliographical references. In the bibliography there are many works that deal with the mechanical characteristics of wood also of other species such as poplar, through destructive and even non-destructive methods. These references, even if not specific to Bamboo, are important because they help the reader to understand and compare the results obtained.

Introduction

Line 33 and in other parts of the text:  

The reference numbers may have changed when they are consecutive in ascending order, for example change "[1], [2], [3], [4], [5]" to [1-5]. There are many references in the text that should be changed as well.

Line 31 and in other parts of the text

The word bamboo appears to be redundant I believe that the authors should modify the text of the manuscript by decreasing the number of times the word "Bamboo" is used. It is clear from the title of what we are talking about in the paper.

Line 78-107:

In this section, the purpose of the work is too generic since we also talk about MOE or flexural failure tests, I suggest the authors to broaden the description of the objectives and to outline them in the final part of the introduction, for example (i) 1 goal (ii) 2 goal etc.

Line 109-113:

I think this paragraph is more appropriate in the conclusions section since we are talking about the results obtained.

Materials and Methods

The description of materials and methods, even if substantial, should be definitely improved still. furthermore, I believe that some vital information for the production of a scientific article is missing, especially the section concerning statistics in this manuscript is not present. I recommend rewriting by improving the entire "materials and methods" section.

Line 136-139

The authors should rewrite this paragraph as there is no mention of the methods of the mechanical tests performed. The authors could add some bibliographical references on the method

Line 136-139

Why do the authors use "pinus silvestris" as a control sample? I think it would be appropriate to use bamboo that is not stressed by biotic to abiotic conditions!

Table 1:

In the table the authors should enter the dendrometric parameters (height, diameter and age)

Line 258-266:

Has the density been calculated according to a regulation? are there any references? I think it is useful to insert references.

Result

The results will be very confusing and do not fully reflect the materials and methods section. in fact, the equations of the modulus of elasticity are present but in the graphs it is not considered, this mechanical property is fundamental for the characteristics of the structural products. I suggest to rewrite the whole section and to carry out statistical analyzes that allow to compare several variables eg: the modulus of elasticity with durability or with chemical characteristics (I think it can give a greater scientific value).

Discussion

There is no discussion section. I recommend writing discussions that can compare your results with others in the bibliography.

Conclusions

Conclusions should be short, so written they resemble a discussion. I suggest the authors rewrite the conclusions by summarizing them and inserting a key to understanding your research and above all future prospects.

Round 2

Reviewer 1 Report

The revised format of the manuscript is well-structured and addresses the comments satisfactorily. Therefore, I believe it is acceptable for publication.

Reviewer 2 Report

General comment:

Thank you for the opportunity to review the manuscript. The authors have definitely improved the manuscript, I appreciate the work of the authors. Only a few fundamental aspects need to be reviewed and addressed to improve the scientific value of this research.

Materials and Methods

I advise the authors to insert a paragraph dedicated to the statistical part as in the results the graphs show statistical relationships or correlations, I think it is correct to specify the statistics adopted in the materials and methods section.

Discussion

The discussions are to the extent exhaustive, I recommend adding the comparisons of some results of this manuscript with similar scientific works by other authors.

Conclusions

Conclusions should be short, so written, again, resemble a discussion. I suggest the authors drop numbers of outcomes, and move comparisons with other studies to the Results and Discussions section.

To Editor

The article has been significantly improved, but some shortcomings remain. The paper may be accepted for publication after minor revisions.
